# Detection of 15-bp Deletion Mutation within *PLAG1* Gene and Its Effects on Growth Traits in Goats

**DOI:** 10.3390/ani11072064

**Published:** 2021-07-10

**Authors:** Zhenyu Wei, Ke Wang, Hui Wu, Zhen Wang, Chuanying Pan, Hong Chen, Xianyong Lan

**Affiliations:** 1Shaanxi Key Laboratory of Molecular Biology for Agriculture, College of Animal Science and Technology, Northwest A&F University, Yangling 712100, China; weizhenyu0222@163.com (Z.W.); lp_wangke@163.com (K.W.); wuhui582@163.com (H.W.); wangzhenid@126.com (Z.W.); panyu1980@126.com (C.P.); chenhong1212@263.net (H.C.); 2Shaanxi Provincial Engineering and Technology Research Center of Cashmere Goats, Yulin University, Yulin 719000, China; 3Life Science Research Center, Yulin University, Yulin 719000, China

**Keywords:** goat, growth traits, *PLAG1* gene, InDel, association

## Abstract

**Simple Summary:**

Goats have always served as an important domesticated livestock. *PLAG1* is a major gene that affects the stature and growth of animals. Body size traits are very important for goats as it directly affects the economic characteristics of meat and cashmere production. This study showed that the 15-base pair (bp) InDel (rs637141549) can significantly affect growth traits such as body weight, height, height at hip cross, chest circumference, hip width and body index of goats through the detection of large samples (*n* = 1581) in four indigenous breeds. Accordingly, it is suggested that the deletion mutation can be used as a potential molecular marker that significantly affects goat growth traits. Moreover, the 15bp deletion mutation can be used as a potential molecular marker, which significantly affects the growth traits of goats and plays an important role in animal husbandry production.

**Abstract:**

Stature and weight are important growth and development traits for animals, which also significantly affect the productivity of livestock. Polymorphic adenoma gene 1 (*PLAG1*) is located in the growth-related quantitative trait nucleotides (QTN), and its variation has been determined to significantly affect the body stature of bovines. This study found that novel 15-bp InDel could significantly influence important growth traits in goats. The frequencies of genotypes of the 15-bp mutation and relationship with core growth traits such as body weight, body height, height at hip cross, chest circumference, hip width and body index were explored in 1581 individuals among 4 Chinese native goat breeds. The most frequent genotypes of Shaanbei white Cashmere goat (SWCG), Inner Mongolia White Cashmere goat (IMCG) and Guanzhong Dairy goat (GZDG) were II genotypes (insertion/insertion), and the frequency of ID genotype (insertion/deletion) was found to be slightly higher than that of II genotype in Hainan Black goat (HNBG), showing that the frequency of the I allele was higher than that of the D allele. In adult goats, there were significant differences between 15-bp variation and body weight, chest circumference and body height traits in SWCG (*p* < 0.05). Furthermore, the locus was also found to be significantly correlated with the body index of HNBG (*p* = 0.044) and hip width in GZDG (*p* = 0.002). In regard to lambs, there were significant differences in height at the hip cross of SWCG (*p* = 0.036) and hip width in IMWC (*p* = 0.005). The corresponding results suggest that the 15-bp InDel mutation of *PLAG1* is associated with the regulation of important growth characteristics of both adult and lamb of goats, which may serve as efficient molecular markers for goat breeding.

## 1. Introduction

Goats (*Capra hircus*) are one of oldest domesticated species and have been used globally for their meat, skins, hair, and milk. According to the Food and Agriculture Organization of the United Nations, there were more than 1.4 billion live goats worldwide in 2011 [1]. Current goat breeding programs are designed for breeding and selection of local dominant species for their excellent characteristics in meat, skin and milk by increasing the growth rate and prolificacy of goats [2,3]. Hence, it is necessary to find effective and practical measures to improve the growth characteristics of goats. Since most growth characters are quantitative with low heritability, it is difficult to improve these traits effectively through traditional phenotypic breeding [4]. However, marker-assisted selection (MAS) is helpful in rapidly establishing a better quality and high-yield population [5,6]. The first step of MAS is to identify some candidate genetic variations. Natural genetic variation can be divided into single nucleotide polymorphism (SNP), insertion/deletion (InDel) and copy number variant (CNV) [7]. Compared with other types of markers, InDel variants possess advantages in convenient detection and conspicuous effects [8,9]. Therefore, InDel has been used to explore the relationship between candidate gene variations and growth traits of different goat breeds, while MAS has been established to provide a theoretical basis for the high-quality and efficient development of the goat industry [10].

China’s goat industry has been domesticated on a large scale for over a millennium. Hainan Black goat (HNBG), Inner Mongolia White Cashmere goat (IMWC), Guanzhong Dairy goat (GZDG) and Shaanbei white cashmere goats (SWCG) are all excellent indigenous breeds with different economic characteristics; however, their economic traits still need improvement. Growth traits have always been the focus of animal breeding and research, especially since many important production traits are related to body size, bringing about huge economic value in agricultural production. As a result, the identification of genetic variations in a candidate gene as well as in body shape related traits of the above four Chinese indigenous breeds is important.

The *PLAG1* gene, with a length of 5.2 kb and four exons, is located on chromosome 14 in goats. It was originally a proto-oncogene found in pleomorphic adenomas of salivary glands [11]. Many studies on human tumors and genetically engineered mouse tumor models have established the importance and versatility of *PLAG1* oncogene in tumorigenesis [12]. *PLAG1* is considered to be one of the main genes in mammalian height, which has a powerful regulatory role in many important growth traits including height. According to the results of numerous genome-wide association studies and meta-analyses, *PLAG1* and various adjacent gene regions have been identified as one of the loci that determine adult human stature both in Asian [13,14,15] and European populations [16,17,18,19]. Meanwhile, *PLAG1* knock-out mice have demonstrated significant growth retardation [20]. In cattle, Karim et al. (2011) determined that the quantitative trait locus (QTL) region located between *PLAG1* and *CHCHD7* had a significant influence on cattle stature and weight. GWAS data has shown that several major loci, including some located in *PLAG1,* explain more than one third of the genetic variations of ten varieties [21]. In recent years, several studies have successively proved that genetic variations in the *PLAG1* gene could regulate cattle weight, body size and other growth traits [22,23,24]. In pigs, *PLAG1* is also considered to be a reliable candidate gene in influencing the function of limb bone length [25,26]. In horses, the *PLAG1* gene variation has been considered to be related to body size [27]. Additionally, in a recent study, researchers have found that mutations in *PLAG1* may be associated with multiple growth traits in sheep [28,29]. Overall, these studies have suggested that the regulatory effect of *PLAG1* on growth traits may be widespread in many mammals. However, few studies exist on the polymorphism of *PLAG1* in goats. Therefore, it is imperative to explore the association of the *PLAG1* gene with goat growth traits.

## 2. Materials and Methods

### 2.1. Animal Sample Collection and DNA Extraction

The experimental goat samples used in this study have been approved by the Animal Use Review Committee of Northwest Agriculture and Forestry University (NWAFU-314020038). The welfare and use of laboratory animals complied with the guidelines of the Ethics Committee. A total of 1581 ear samples were gathered from healthy goats of the following Chinese native goat breeds: Hainan Black goat (HNBG, *n* = 212, all adults), Guanzhong Dairy goat (GZDG, *n* = 91, all adults), Inner Mongolia White Cashmere goat (IMWC, *n* = 452, including 69 adults and 383 lambs) and Shaanbei white Cashmere goat (SWCG, *n* = 826, including 737 adults and 89 lambs). The main growth trait statistics of the four indigenous breeds are shown in Table 1 (to ensure the accuracy of the data analysis, some phenotypic data that may be measured inaccurately were discarded or removed, which did not affect the detection of their genotype data and the analysis of population genetic parameters). The genomic DNA of all SWCG and IMCG individuals were then extracted from the ear tissues of animals raised in the Yulin goat farm in Shaanxi Province, which were all recorded growth trait data. GZDG samples were collected from the breeding base in Fuping, Shaanxi. The HNBGs were raised in Hainan Province. The growth-related data of these goats were measured and collated by veterinarians of these farms, including body weight (BW, kg), body length (BL, cm), chest circumference (CC, cm), body height (BH, cm), height at hip cross (HAH, cm), hip width (HW, cm), chest depth (CD, cm), chest width (CW, cm), circumference of the cannon bone (CCB, cm), body index(BI, $\mathrm{BI}=\frac{\mathrm{CC}}{\mathrm{BL}}\times 100\%$), chest circumference index (ChCI, $\mathrm{ChCI}=\frac{\mathrm{CC}}{\mathrm{BH}}\times 100\%$) and body length index (BLI, $\mathrm{BLI}=\frac{\mathrm{BL}}{\mathrm{BH}}\times 100\%$) [28].

Genomic DNA was obtained using high-salt extraction [29,30]. Following an analysis using an ultra-micro spectrophotometer, all DNA samples were diluted to 20 ng/μL of the same standard solution and stored temporarily at 4 °C. In addition, 30 individual DNA samples were mixed into a genomic DNA pool for analysis via polymerase chain reaction (PCR) [30].

### 2.2. Primer Design, InDel Genotyping and DNA Sequencing

Using the Ensemble database (http://www.ensembl.org/index.html, accessed on 1 October 2018) and the goat *PLAG1* gene sequence (version: ENSCHIG00000000623.1), four pairs of primer for producing the 15 bp InDel locus (NC_030821.1:rs637141549, g.58752100_58752114delGAGGAGGGAGGGTTT ) in the downstream were designed (Table 2). Assays were then measured via touch-down PCR containing 8 ng genomic DNA [10]. The PCR products were separated by 2% agarose gel electrophoresis with the aim of genotyping InDel polymorphisms in goat *PLAG1*. Sequencing the PCR products of different genotypes were subsequently performed in order to verify the mutations [31]. The sequencing samples were homozygous wild and mutant genotypes of SWCG. PCR products of the homozygous wild type (insertion/insertion: II) consisted of a single fragment, while that of the heterozygous mutant (insertion/deletion: ID) consisted of two fragments. Meanwhile, the PCR product of the homozygous mutant (deletion/deletion: DD) consisted of a single fragment.

### 2.3. Statistical Analysis

The method provided by Botstein was used to calculate the genotype and allele frequencies of the InDel mutation in the goat breeds, and the genetic structure of the four native goat populations was investigated [32]. Meanwhile, the SHEsis program (http://analysis.bio-x.cn, accessed on 1 October 2018) was used to analyze the Hardy-Weinberg equilibrium (HWE) structure of the locus in the four native breeds [8,33]. The population genetic diversity indices were then calculated using Nei’s method [34], in which the degree of homozygosity (Ho) and heterozygosity (He) (Ho + He = 1) were the measures of the genetic variation of the population. Polymorphism information content (PIC) is an indicator of population polymorphism [32].

The statistical linear model: Y_ij_ = μ + G_i_ +e_ij_, where Y_ij_ represents the observation of growth-related traits; μ refers to the overall average of each trait; G_i_ represents the fixed effect of genotype or combined genotype; and e_ij_ stands for random residuals [35,36]. The fixed effect of the genotype was the main reason for the difference in growth traits, in which the difference was considered to be significant if the *p* value was less than 0.05 [28]. The independent sample t-test and one-way analysis of variance with post-hoc test was used to evaluate the association between the goat *PLAG1* gene and main growth traits in SPSS (Version 24.0, Inc., Chicago, IL, USA).

## 3. Results

### 3.1. Detection and Genotyping of the PLAG1 Gene in Goat

Combined with DNA sequencing and alignment analysis, one 15-bp InDel (InDel NC_030821.1:rs637141549, g.46441-46455delGAGGAGGGAGGGTTT, P4-15-bp InDel) in downstream of the *PLAG1* gene was identified in all four breeds (Figure 1). Genotyping of *PLAG1* gene InDel was performed by PCR-based amplification of fragment length polymorphism and three different genotypes of the locus could be obtained: homozygote insertion type (II, 205 bp), deletion type (DD, 190 bp), and heterozygote type (ID, 205 bp and 190 bp). The proof results were verified by PCR products sequencing, the deletion sequence is GAGGAGGGAGGGTTT. Based on these results, the locus is consistent with the predicted InDel information of the Ensemble database.

### 3.2. Population Genetic Parameter Analysis

The genotype distribution, genotype frequencies and allele frequencies (Ho, He, Ne and PIC) of the *PLAG1* genes among the four breeds goat are shown in Table 3. For P1-15-bp InDel, the most frequently encountered genotypes of SWCG, IMCG and GZDG were all found to be II genotypes. However, in HNBG, the frequency of the ID genotype was slightly higher than that of the II genotype. Specifically, the frequency of the D allele was lower than that of the I allele, suggesting that I was the dominant allele. The experimental results demonstrated that the Ne of all breeds was between 1 and 2. Meanwhile, the 15-bp InDel showed intermediate polymorphism (0.25< PIC< 0.5) in HNBG. In addition, all three varieties exhibited low polymorphism. These results indicated that the InDel mutation of goat *PLAG1* contained rich genetic diversity in the analyzed population. Moreover, the χ2 test indicated that none of the tested populations met HWE (*p <* 0.05). Accordingly, it was clear that feeding management factors significantly affected the distribution of genotype and allele frequency.

### 3.3. Relationship between PLAG1 InDel and Growth-Related Traits

Different genotypes of 15 bp- Indel mutations were found to be significantly related to the core growth characteristics in goat breeds, with body weight, body height, height at hip cross, chest circumference, hip width and body index of four superior varieties (*p* < 0.05 or *p* < 0.01) (Table 4 and Table 5). Among them, extremely significant differences were present in the body weight traits in adult SWCG (*p* = 0.0006). Moreover, significant differences in body height of adult SWCG (*p* = 0.022) existed, while there were significant differences in height at hip cross of lamb SWCG (*p* = 0.036). Additionally, different genotypes of this variation locus were noted to be significantly correlated with the chest circumference of SWCG and body index of HNBG (*p* = 0.044). Interestingly, the II genotype was found to be both a dominant and superior genotype, having a better phenotype than the ID genotype, regardless of breed and age. However, in the other two breeds, the ID genotype had an extremely significant difference in hip width in adult GZDG (*p* = 0.002) and lamb IMWC (*p* = 0.005).

## 4. Discussion

Genetic resources serve as strategic resources in agricultural production. In previous studies, gene variations have been shown to be associated with the growth-related traits of goats [28,37,38,39]. In goat breeding and husbandry, growth traits play a vital role in a variety of economic characteristics, such as meat and fur production; hence, exploring candidate molecular makers for goat breeding is important. In this study, *PLAG1* was identified as a candidate gene affecting goat growth traits, and the relationship between the 15 bp InDel (rs637141549) and the growth traits of a different breed of goat was observed. The analysis of genetic parameters showed that the 15 bp InDel was not at Hardy-Weinberg equilibrium state in the four breeds, which may have been due to artificial selection, genetic drift and migration causing mutations in the locus [40]. Additionally, the allele frequency of “I” was noted to be higher than that of “D” in all goat breeds. Compared to the DD genotype, II was the dominant genotype in SWCG and had a distinctly better phenotype, while the phenotypes of GZDG and IMWC were in reverse. Here, rs637141549 (15 bp InDel) of *PLAG1* was found to be significantly correlated with body weight (*p* = 0.0006), body height (*p* = 0.022), chest circumference (*p* = 0.033), hip width (*p* = 0.002) and body index (*p* = 0.044) of adults in different goat breeds. In lambs, significant correlations were present between this variation with growth traits, including height at hip cross (*p* = 0.036) and hip width (*p* = 0.005). Evidently, the phenotypes of each breed were positively correlated. Hence, it was further speculated that rs637141549 (15 bp InDel) may also affect other growth-related phenotypes of goat breeds. 

*PLAG1* is a millennia-old derived allele, which has an important effect on body weight and body size [41]. As a transcription factor, the N-terminus of *PLAG1* contains seven canonical C2H2 zinc finger domains responsible for the DNA-binding ability of proteins, while the C-terminus contains a serine-rich region that has transcriptional activation activity. At the same time, it can be used as a nucleoprotein and participates in the docking between protein and nuclear pores through the N-terminal zinc finger region as well as the nucleoprotein A2 recognition site [12]. However, evidence pertaining to an important role of *PLAG1* in mammalian growth has been increasing, though its specific mechanism requires further elucidation. For example, it is clear that *PLAG1* highly upregulated the expression levels of Insulin-like growth factor 2 (*IGF2*) and other important growth factors. *IGF2* plays an essential role in bone growth and development, myogenesis, vasculogenesis, and cardiac development [42,43,44]. Moreover, *PLAG1* could bind the *IGF2* promoter and stimulate its activity, illustrating that *PLAG1* plays an indispensable role in promoting its transcriptional expression [45]. In addition, Van Laere et al. identified a mutation located in QTL of the *IGF2* gene that affects pig muscle growth (*IGF2*-intron3-G3072A), indicating that regulatory mutation has an important role in controlling phenotypic variation. This was a typical example of non-coding region variation that may influence transcription efficiency and mRNA stability [46,47,48,49].

Simultaneously, *PLAG1* has been found to induce the upregulation of several genes associated with Wnt signaling, which are essential for the growth and development of mice [50]. Moreover, studies have found that the *CHCHD7* gene shares a bidirectional promoter with *PLAG1*, which is only 440-bp away from *PLAG1*. As a fusion partner of *PLAG1*, *CHCHD7* mutation has recently been shown to regulate the growth traits of goats [51]. Similarly, as the upstream activator of *PLAG1*, *HMGA2* can also affect the body height of adults and lambs. Accordingly, it is speculated that these results may relate to *PLAG1*, resulting in the regulation of body height and growth traits [52]. This was consistent with the result in which *PLAG1* knockout mice showed dwarfism without other symptoms, while cattle with *PLAG1* gene overexpression increased in withers height. In addition, *PLAG1* expression was different in early life and adulthood [22,53]. These results were identified between this mutation and the growth traits among the four goat breeds at different growth stages in order to ensure accurate and reliable results. The results obtained in this study shows that *PLAG1* mutation has similar effects on growth and development in goats as in other mammals. Briefly, the 15-bp InDel mutation of *PLAG1* is of great value in the genetic selection of growth traits in goats and may serve as an effective molecular marker in selection of goats with better growth performance. *PLAG1* may be one of the most promising candidate genes for the breeding of goats, and the regulatory mutations within *PLAG1* are important for selecting the corresponding phenotypic traits [54,55].

## 5. Conclusions

The 15-bp deletion mutation of *PLAG1* was detected in large samples among four native superior goat breeds. The results demonstrated that it can significantly improve growth traits, such as stature and weight. These results indicate that 15-bp InDel variation of *PLAG1* is of great value in the genetic selection of growth traits in goats and may serve as a useful DNA marker for selecting excellent goat individuals.

## Figures and Tables

**Figure 1 animals-11-02064-f001:**
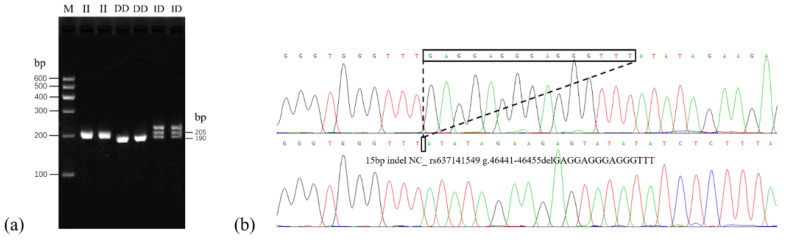
(**a**) Agarose gel electrophoresis patterns of goat PLAG1 gene P4-15-bp InDel. Maker, M. (**b**) Sequencing maps for the 15 bp InDel within the goat PLAG1 gene. The sequence within the black border shows the difference in the sequence fragment.

**Table 1 animals-11-02064-t001:** Descriptive statistics of the main growth traits of four native breeds.

Breeds	Growth Traits	Effective Sample Size	Mean (cm)	Standard Error	Minimum (cm)	Maximum (cm)
HNBG	BH	205	52.80	0.28	43.20	63.20
	BL	205	56.15	0.31	46.00	68.30
	CC	205	72.38	0.46	54.00	91.00
	CD	205	26.56	0.16	19.20	33.00
	CW	205	14.91	0.13	10.80	21.60
	HW	205	13.71	0.94	10.20	17.20
	CCB	204	7.80	0.51	6.00	10.00
GZDG	BH	80	73.53	0.48	63.00	83.00
	BL	80	74.81	0.49	65.00	86.00
	HAH	80	74.08	0.47	64.00	84.00
	CC	26	87.50	0.48	83.00	92.00
	CW	26	16.69	0.09	16.00	17.00
	CCB	26	8.00	0.00	8.00	8.00
SWCG	BW	649	48.91	0.61	15.00	79.00
	BH	812	56.02	0.18	36.50	73.00
	BL	634	66.05	0.24	46.00	83.00
	HAH	811	58.94	0.17	23.20	73.00
	CC	814	84.61	0.30	62.00	114.00
	CD	813	29.00	0.11	14.70	42.00
	CW	814	19.91	0.13	11.50	34.00
	HW	472	15.40	0.09	10.00	21.00
	CCB	725	8.31	0.09	5.60	11.00
IMWC	BW	452	24.13	0.36	11.00	54.80
	BH	450	51.44	0.20	39.50	65.00
	BL	451	56.53	0.31	41.00	75.00
	HAH	450	53.29	0.23	43.50	68.00
	CC	450	69.61	0.37	55.50	97.00
	CD	451	24.27	0.16	16.00	38.50
	CW	451	14.74	0.13	9.00	24.00
	HW	450	12.97	0.12	8.50	53.50
	CCB	450	8.43	0.03	7.00	10.00

Body height, BH; body length, BL; chest width, CW; chest depth, CD; chest circumference, CC; height across the hip, HAH; rump length, RL; hip width, HW; circumference of cannon bone, CCB.

**Table 2 animals-11-02064-t002:** PCR primers used for detecting InDel loci of goat *PLAG1* gene.

Primer	Primer sequences (from 5’ to 3’)	Length	Region	Tm*
P1	F: GGGGCGTCAACCTAAGAAACAT R: GAAATCCGCAGTGATAGCTGG	210	Intron 1	TD
P2	F: ATGAGTTACTGGTAAGGAGA R: AGAAGGCAATGGCAC	209	Intron 1	TD
P3	F: GAGATTCCTGTTAGAATTTATAGT R: CTGAAAAGCCCCGTACT	132	Intron 1	TD
P4	F: TGAGCAACAGGGAGGGTA R: TGGTGGCTACATCCAAGC	205	Downstream	62 °C

F: forward; R: reverse; Tm*: Temperature parameters for primer and template binding, when 50% of the primer and complementary sequence appear as double-stranded DNA molecules. TD: touch-down PCR protocol, temperatures of anneal is during 50–68 °C.

**Table 3 animals-11-02064-t003:** Genotypic, allelic frequencies, and population indexes for the15-bp InDel in four indigenous goat breeds.

Breeds	Size	Genotypic Frequencies			Allelic Frequencies		HWE	Population Parameters			
	N	II	ID	DD	I	D	*p* Values	Ho	He	Ne	PIC
SWCG	826	0.805	0.190	0.005	0.900	0.100	*p* < 0.05	0.820	0.180	1.219	0.163
IMWC	452	0.695	0.281	0.024	0.835	0.165	*p* < 0.05	0.725	0.275	1.380	0.237
HNBG	212	0.439	0.486	0.075	0.682	0.318	*p* < 0.05	0.566	0.434	1.767	0.340
GZDG	91	0.944	0.056	0.000	0.973	0.027	*p* < 0.05	0.947	0.053	1.056	0.052

SWCG, Shaanbei White Cashmere goat; IMWC: Inner Mongolia White Cashmere goat; HNBG: Hainan Black goat; GZDG: Guanzhong Dairy goat; HWE, Hardy–Weinberg equilibrium; Ho, observed homozygosity; He, heterozygosity; Ne, effective allele numbers; PIC, Polymorphism information content.

**Table 4 animals-11-02064-t004:** Association of the 15-bp InDel and growth traits of four adult indigenous goat breeds (mean ± SE).

Breeds	Growth Traits	Genotypes			*p* Value
		II	ID	DD	
SWCG (*n* = 683)	**BW (kg** **)**	**53.23 ± 0.73 ^A^ (*n* = 436)**	**46.79 ± 1.39 ^B^ (*n* = 120)**	—	**0.0006**
	**BH (cm)**	**57.01 ± 0.20 ^a^ (*n* = 544)**	**55.96 ± 0.46 ^B^ (*n* = 136)**	—	**0.022**
	BL (cm)	67.10 ± 0.30 ^b^ (*n* = 392)	66.54 ± 0.53 ^a^ (*n* = 111)	—	0.369
	HAH (cm)	19.87 ± 0.17 (*n* = 545)	20.32 ± 0.34 (*n* = 137)	—	0.223
	**CC (cm)**	**59.83 ± 0.18 ^a^ (*n* = 542)**	**58.69 ± 0.49 ^b^ (*n* = 137)**	—	**0.033**
	CD (cm)	15.45 ± 0.12 (*n* = 285)	15.45 ± 0.21 (*n* = 95)	—	0.982
	CW (cm)	8.34 ± 0.12 (*n* = 545)	8.31 ± 0.07 (*n* = 138)	—	0.899
	HW (cm)	85.08 ± 0.36 (*n* = 544)	83.87 ± 0.74 (*n* = 138)	—	0.135
	CCB (cm)	29.20 ± 0.13 (*n* = 544)	29.2 ± 0.23 (*n* = 137)	—	0.862
HNBG (*n* = 205)	BH (cm)	52.84 ± 0.43 (*n* = 92)	52.87 ± 0.41 (*n* = 98)	52.05 ± 1.11 (*n* = 15)	0.749
	BL (cm)	55.96 ± 0.44 (*n* = 92)	56.51 ± 0.48 (*n* = 98)	55.05 ± 1.10 (*n* = 15)	0.383
	CC (cm)	72.88 ± 0.62 (*n* = 92)	72.01 ± 0.69 (*n* = 98)	71.80 ± 2.08 (*n* = 15)	0.652
	CD (cm)	26.71 ± 0.21 (*n* = 92)	26.37 ± 0.25 (*n* = 98)	26.85 ± 0.16 (*n* = 15)	0.567
	CW (cm)	15.09 ± 0.17 (*n* = 92)	14.7 ± 0.21 (*n* = 98)	14.9 ± 0.13 (*n* = 15)	0.517
	HW (cm)	13.78 ± 0.14 (*n* = 92)	13.69 ± 0.13 (*n* = 98)	13.5 ± 0.46 (*n* = 15)	0.670
	CCB (cm)	7.82 ± 0.74 (*n* = 92)	7.76 ± 0.73 (*n* = 97)	7.90 ± 0.24 (*n* = 15)	0.722
	**BI**	**130.68 ± 1.07 ^a^ (*n* = 92)**	**127.52 ± 0.71 ^b^ (*n* = 98)**	**130.36 ± 2.59 ^a^ (*n* = 15)**	**0.044**
	BLI	106.22 ± 0.84 (*n* = 92)	107.09 ± 0.77 (*n* = 98)	106.19 ± 0.55 (*n* = 15)	0.687
	ChCI	138.19 ± 1.00 (*n* = 92)	136.33 ± 0.98 (*n* = 98)	138.15 ± 3.50 (*n* = 15)	0.412
GZDG (*n* = 80)	BH (cm)	73.35 ± 0.51 (*n* = 75)	76.2 ± 1.07 (*n* = 5)	--	0.147
	BL (cm)	74.63 ± 0.51 (*n* = 75)	77.60 ± 1.40 (*n* = 5)	**--**	0.132
	HAH (cm)	73.92 ± 0.49 (*n* = 75)	76.40 ± 1.36 (*n* = 5)	--	0.197
	**HW (cm)**	**16.65 ± 0.10 ^B^ (*n* = 23)**	**17.00 ± 0.00 ^A^ (*n* = 3)**	--	**0.002**
	CC (cm)	87.35 ± 0.51 (*n* = 23)	88.67 ± 1.67 (*n* = 3)	--	0.395
	CCB (cm)	8.00 ± 0.00 (*n* = 23)	8.00 ± 0.00 (*n* = 3)	--	0.000
IMWC (*n* = 68)	BW (kg)	34.30 ± 1.06 (*n* = 47)	36.74 ± 1.66 (*n* = 21)	--	0.211
	BH (cm)	56.86 ± 0.51 (*n* = 47)	55.81 ± 0.76 (*n* = 21)	--	0.255
	BL (cm)	66.67 ± 0.62 (*n* = 47)	66.40 ± 0.80 (*n* = 21)	--	0.805
	HAH (cm)	59.86 ± 0.56 (*n* = 47)	60.35 ± 0.77 (*n* = 21)	--	0.615
	CC (cm)	78.48 ± 0.94 (*n* = 47)	81.02 ± 1.48 (*n* = 21)	--	0.142
	CD (cm)	29.52 ± 0.33 (*n* = 47)	20.40 ± 0.44 (*n* = 21)	--	0.841
	CW (cm)	18.62 ± 0.37 (*n* = 47)	19.21 ± 0.52 (*n* = 21)	--	0.361
	HW (cm)	15.97 ± 0.83 (*n* = 47)	15.36 ± 0.23 (*n* = 21)	--	0.611
	CCB (cm)	8.38 ± 0.68 (*n* = 47)	8.57 ± 0.11 (*n* = 21)	--	0.140

Body weight, BW; body height, BH; body length, BL; chest width, CW; chest depth, CD; chest circumference, CC; height across the hip, HAH; rump length, RL; hip width, HW; circumference of cannon bone, CCB; body index, BI; body length index, BLI; chest circumference index, ChCI. The traits that have a significant association between genotype and phenotype are bold (^A,B^ *p <* 0.01; ^a,b^: *p <* 0.05).

**Table 5 animals-11-02064-t005:** Association of the 15-bp InDel and growth traits in lambs of SWCG and IMWC goat breeds (mean ± SE).

Breeds	Growth Traits	Genotypes			*p* Value
		II	ID	DD	
SWCG (*n* = 89)	BW (kg)	35.67 ± 0.90 (*n* = 73)	32.91 ± 1.74 (*n* = 16)	—	0.187
	BH (cm)	49.99 ± 0.44 (*n* = 73)	48.89 ± 1.10 (*n* = 16)	—	0.305
	BL (cm)	62.90±0.62 (*n*=73)	62.29 ± 1.47 (*n* = 16)	—	0.681
	**HAH (cm)**	**54.03 ± 0.50 ^a^ (*n* = 73)**	**51.46 ± 1.26 ^b^ (*n* = 16)**	—	**0.036**
	CC (cm)	79.85 ± 0.84 (*n* = 73)	76.81 ± 1.85 (*n* = 16)	—	0.130
	CD (cm)	28.19 ± 0.29 (*n* = 73)	27.45 ± 1.05 (*n* = 16)	—	0.345
	CW (cm)	20.57 ± 0.32 (*n* = 73)	20.17 ± 0.75 (*n* = 16)	—	0.602
	HW (cm)	15.20 ± 0.20 (*n* = 73)	14.88 ± 1.69 (*n* = 16)	—	0.493
	CCB (cm)	8.01 ± 0.10 (*n* = 73)	7.62 ± 0.24 (*n* = 16)	—	0.122
IMWC (*n* = 373)	BW (kg)	21.79 ± 0.34 (*n* = 267)	22.94 ± 0.59 (*n* = 106)	—	0.076
	BH (cm)	50.69 ± 0.23 (*n* = 265)	50.18 ± 0.40 (*n* = 106)	—	0.245
	BL (cm)	54.56 ± 0.33 (*n* = 266)	55.18 ± 0.49 (*n* = 106)	—	0.304
	HAH (cm)	52.19 ± 0.24 (*n* = 265)	52.01 ± 0.40 (*n* = 106)	—	0.692
	CC (cm)	67.46 ± 0.39 (*n* = 265)	68.71 ± 0.63 (*n* = 106)	—	0.089
	CD (cm)	23.30 ± 0.17 (*n* = 266)	23.37 ± 0.25 (*n* = 106)	—	0.828
	CW (cm)	13.90 ± 0.13 (*n* = 266)	14.30 ± 0.21 (*n* = 106)	—	0.105
	**HW (cm)**	**12.34 ± 0.10 ^B^(*n* = 266)**	**12.86 ± 0.15 ^A^ (*n* = 105)**	—	**0.005**
	CCB (cm)	8.43 ± 0.03 (*n* = 265)	8.42 ± 0.06 (*n* = 106)	—	0.977

**Note**: body height, BH; body length, BL; chest width, CW; chest depth, CD; chest circumference, CC; height across the hip, HAH; rump length, RL; hip width, HW; circumference of cannon bone, CCB; BLI, body length index; ChCI, chest circumference index. The traits that have a significant association between genotype and phenotype are bold (^A,B^ *p <* 0.01; ^a,b^ *p <* 0.05).

## Data Availability

Data is contained within the article.

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
