# Peer review of "Detection of 15-bp Deletion Mutation within PLAG1 Gene and Its Effects on Growth Traits in Goats"

_animals, 2021, doi:10.3390/ani11072064_

Round 1

Reviewer 1 Report

A minor revision is suggested by considering the recommendations below for the manuscript entitled “Detection of 15-bp deletion mutation within PLAG1 gene and 2 its effects on growth traits in goats”. The scope of the research is quite valuable regarding the possible contribution to the literature. However, for the critical set of information created by the authors to effectively have an impact on the relevant literature, it is important to revise what are listed below.

Introduction:

The introduction has summarized the problem sufficiently for the study to be understood. It was found enough.

Materials and methods:

In the introduction part, it is mentioned that 4 different goat breeds used in the study have different economic characteristics. however, the descriptive statistics of the animals in terms of the phenotypes of interest are not specified. Descriptive statistics (mean, max., min., coefficient variation, etc.) of the interested traits regarding different breeds should be presented in a table.

It will be more explanatory to give short descriptions of the traits (especially body index-BI, chest circumference index-ChCI and body length index-BLI) in the text.

Line 143. How were the breeds included in the model? Were all animals analysed in one model? if all animals were analysed in a model, how was the effect of breed put into the model? Results for alleles which are fixed for a breed will be false positive, when the breed effect is not included in the model. Please clarify the breed effect.

Results:

Line 176. Please give more detail, how the feeding management factors effect HWE in the populations? Give detailed explanation in the perspective of the results.

General:

Throughout the paper, fluency struggles due to the unclear sentences, problematic word choice in key sentences as well as collocation problems. Therefore, a revision of the text is strongly suggested especially for the introduction part.

Author Response

I have made a detailed response to your suggestions and questions in the attachment, I hope these can meet your requirements, thank you!

Reviewer 2 Report

The authors investigated an association of the 15-bp InDel mutation of PLAG1  with the regulation of important growth characteristics of both adult and lamb goats. The manuscript contains some novel information, however, it requires extensive editing of the English language and style and some minor improvement before publication.

line 30- please explain the abbreviation

line38 - correct to "provide"

line 43- Latin names should be written in italics

line 46- correct to "programs"

line 97- Do the animals were randomly selected for the study?

line 120 - change isolated to "separated in .."

line 121- 122-  please rearrange this sentence. "Select..."

Table 3 - why the number of goats eg. SWCG is 683 but in the text is 737 adults. When I summarise the number of animals with different genotype results are different numbers. please explain. If authors do not include some animals please add this information in the text.

table 4- unreadable- row numbers merged with the table

line 264-263- It seems like it is rather an introduction than a disciusion. 

Please add some more information about indel PLAG1 gene studies in other species eg. sheep. If there is no research on goats you should try to compare your result with different species in the discussion.

https://www.tandfonline.com/doi/abs/10.1080/10495398.2021.1906265?journalCode=labt20

https://www.sciencedirect.com/science/article/pii/S0921448819301385

Author Response

(The authors gave the same response as above.)

Reviewer 3 Report

Manuscript “Detection of 15-bp deletion mutation within PLAG1 gene and its effects on growth traits in goats” provides some information regarding the association of PLAG1 with growth traits in four Chinese goat breeds. Overall, the manuscript is acceptable, but the statistical section is not clear (how the authors using the ages in the models and why the authors tested the correlation but not the association). The authors should edit the section or re-analyze the association using the proper models and effects.

Comments

Line 18: Remove “excellent” and add an exact number of breeds.

Line 21: Which plays an important role in animal husbandry production, goats, or markers. Please write the sentence clearly.

Line 25: define QTN before using it as an abbreviation.

Line 29: Replace excellent to Chinese to specify the countries of indigenous breeds.

Line 30: Write full names for SWCG, IMCG, and GZDG

What are II genotypes? It is not clear given the current information from the abstract; the authors need to specify it.

slightly higher: is it significant?

Line 32: Remove extremely and remove non-scientific statements through the manuscript.

There is no information to know the changes between different genotypic groups.

Line 46: Why the data in 2011, the authors can update the new data 2020.

Line 52-53: MAS is not useful for animal breeding; genomic selection is.

Line 69: change was to is

Line 71-72: Is the manuscript involving cancer?
The gene names should be in Italics.

Line 70-90: The authors should add some information about the gene structure (length, numbers of exons, introns, etc. in goats)

Line 138: What did the authors mean “cultivars”?
Line 148: Change T test to t-test

Line 149: Add the versions of SPSS?
Line 153: The letter I should be removed.

Line 259: Change What's more, to In addition.

Line 143-144: Are these goats all males or females? If not the Sex should be included in the model. Did they have any relationship with each other (siblings or half-sibs, etc.)

How did the authors include the age of animals: as the fixed effects (so how many levels in the model) or covariate?

Did the authors sequence the PCR product to confirm the sequence of the DNA?

Line 147-149: The statistical sections are not clear; the authors should not test the correlation but should run the association analyses. The effects of genotypes could be done using a post-hoc test.

Author Response

(The authors gave the same response as above.)

Round 2

Reviewer 2 Report

Dear Authors

All my previous suggestions have been introduced into the manuscript. Now, I am content with the new version of this manuscript.

Kind regards 

Reviewer

Author Response

Dear reviewer,

I am very glad that you have put forward useful suggestions in this article. We will continue to work hard to make our research more rigorous and valuable. Thank you again for your affirmation and help!

Kind regards,

The first author (Zhenyu Wei)

Corresponding author (Prof. Xianyong Lan)

Corresponding address: College of Animal Science and Technology, Northwest A&F University, Shaanxi Key Laboratory of Molecular Biology for Agriculture, Yangling, Shaanxi, 712100, China

Reviewer 3 Report

The authors have been addressed all my comments. I have some minor suggestions:

Please check if all gene names are in Italics 

Line 476: Add the reference and version of Ensembl gene databases.

Line 585: The authors could use abbreviations here.

Line 693: If the authors want to use P values as Italics, then using them consistently 

Table 4 and 5: The authors might remove (mean + SE) as it is not necessary for the table titles. 

Define bp as base pairs before using it as an abbreviation 

Line 970:   might change "selecting excellent goat individuals" to the selection of goats with better growth performance. 

Line 971: remove excellent

For Figure 1. The authors might add a footnote to explain the band and M as markers as well as the sequencing information.  Moreover, the authors might separate them as A for gel and B for sequencing pictures, respectively. 

Author Response

Thanks for your useful suggestion. I have made a detailed response to your suggestions and questions in the attachment, I hope these can meet your requirements, thank you!
